



# **Moist moss tundra on Kapp Linne, Svalbard is a net source of $CO_2$ and $CH_4$ to the**
# **atmosphere**
**Anders Lindroth[1], Norbert Pirk[2], Ingibjörg S Jónsdóttir[3], Christian Stiegler[4], Leif**
**Klemedtsson[5], and Mats B Nilsson[6]**
[1]Department of Physical Geography and Ecosystem Science, Lund University, Lund, Sweden.
[2]Department of Geosciences, University of Oslo, Oslo, Norway.
[3]Life and Environmental Sciences, University of Iceland, Reykjavik, Iceland.
[4]Bioclimatology, Georg-August Universität Göttingen, Göttingen, Germany.
[5]Department of Earth Sciences, University of Gothenburg, Gothenburg, Sweden.
[6]Department of Forest Ecology and Management, Swedish University of Agricultural Sciences,
Umeå, Sweden.
Corresponding author: anders.lindroth@nateko.lu.se



**Abstract**

We measured $CO_2$ and $CH_4$ fluxes using chambers and eddy covariance (only $CO_2$) from a moist moss tundra in Svalbard. The average net ecosystem exchange (NEE) during the summer (June-August) was -0.40 g C m$^{-2}$ day$^{-1}$ or -37 g C m$^{-2}$ for the whole summer. Including spring and autumn periods the NEE was reduced to -6.8 g C m$^{-2}$ and the annual NEE became positive, 24.7 g C m$^{-2}$ due to the losses during the winter. The $CH_4$ flux during the summer period showed a large spatial and temporal variability. The mean value of all 214 samples was 0.000511±0.000315 µmol m$^{-2}$s$^{-1}$ which corresponds to a growing season estimate of 0.04 to 0.16 g $CH_4$ m$^{-2}$. We find that this moss tundra emits about 94-100 g $CO_2$-equivalents m$^{-2}$ yr$^{-1}$ of which $CH_4$ is responsible for 3.5-9.3% using GWP$_{100}$ of 27.9 respectively GWP$_{20}$.

Air temperature, soil moisture and greenness index contributed significantly to explain the variation in ecosystem respiration ($R_{eco}$) while active layer depth, soil moisture and greenness index were the variables that best explained $CH_4$ emissions. Estimate of temperature sensitivity of $R_{eco}$ and gross primary productivity showed that a modest increase in air temperature of 1 degree did not significantly change the NEE during the growing season but that the annual NEE would be even more positive adding another 8.5 g C m$^{-2}$ to the atmosphere. We tentatively suggest that the warming of the Arctic that has already taken place is partly responsible for the fact that the moist moss tundra now is a source of $CO_2$ to the atmosphere.

**1 Introduction**

Climate warming is predicted to be most evident at high latitudes (Friedlingstein et al., 2006) with profound effects on ecosystem functioning. One of the high latitude regions that are expected to experience the most dramatic changes caused by climate change is the Arctic. This region which is located roughly north of the tree-line is characterized by cold winters and cool summers and with mean annual temperatures below zero. The summer periods are short ranging between 3.5 to 1.5 months from the southern boundary to the north and July is normally the warmest month. Annual precipitation is generally low decreasing from about 250 mm in the southern areas to 45 mm in polar deserts in the north (Callaghan et al., 2005).

The permafrost soils in the Arctic store 1035±150 Pg of organic carbon in the top 0-3 m (Hugelius et al., 2014) which is more than the average 2010-2019 of 860 Pg of carbon in the atmosphere (Friedlingstein et al., 2020). The increased warming in these areas can induce higher decomposition rates due to increased microbial activity which will provide a positive feedback to the climate system (Schuur et al., 2015). On the other hand, warming can also increase photosynthesis and carbon uptake and thus compensate for, or exceed, the effect of increased decomposition. Climate warming is also affecting plant community composition and the length of the growing season (Post et al., 2009) which also has an impact on the processes regulating annual carbon emissions and uptake (Bosiö et al., 2014). There is however a large uncertainty regarding the timing, magnitude and possible sign of potential feedbacks caused by these changes (Myers-Smith et al., 2020).

Understanding processes that are controlling the exchanges of greenhouse gases in the Arctic is crucial for assessment of potential feedback effects. For this purpose, multiple year-around long-term studies including direct measurements of $CO_2$ and $CH_4$ fluxes covering all seasons, winter, spring, summer and autumn would be ideal. This is a great challenge in the harsh climate of the



Arctic and with limited support of key infrastructures for, e.g., provision of electricity for
operation of instruments.
In spite of these difficulties a few year-around studies have been performed during the last
couple of decades. In the low Arctic, Oechel et al. (2013) demonstrate the importance of the
wintertime fluxes in a tussock tundra ecosystem in Alaska. They found that the non-summer
season emitted more $CO_2$ than the corresponding uptake during the summer resulting in a net
source to the atmosphere of about 14 g C m$^{-2}$ on an annual basis. They also showed that the
shoulder seasons, spring and autumn roughly out-weighted the summer uptake. Euskirchen et al.
(2012, 2016) measured net $CO_2$ exchange in three different tundra ecosystems; heath tundra,
tussock tundra and wet sedge tundra in northern Alaska over three years. They found that the
uptake of -51 to -95 g C m$^{-2}$ during the summer (June-August) was overturned by the respiration
that occurred during the winter period resulting in net annual losses for all three ecosystems.
Zhang et al. (2019) reported five years of year-around flux measurements in a heath ecosystem
on west Greenland and they found that the heath was an annual sink of -35±15 g C m$^{-2}$. One year
with an anomalously deep snow pack showed a 3-fold higher respiration during the winter as
compared to the other years which resulted in a significantly lower net uptake during that year.
Even fewer studies have been done on year-roud studies in the high Arctic. Lüers et al. (2014)
quantified the annual $CO_2$ budget using eddy covariance measurements in a river catchment area
near Ny-Ålesund on Spitsbergen in the Svalbard archipelago and they found that the ecosystem
was in C-balance. The footprint area was a semi-polar desert with only 60% vegetation cover and
patches of bare soil and stones. Also in Svalbard but further south in Adventdalen on a flat
alluvial fen irregularly covered with ice wedged polygons, Pirk et al. (2017) made year-around
measurements of $CO_2$ fluxes and found it to be a net sink of -82 g C m$^{-2}$. Because of the
irregularities caused by the ice wedges and the differences in wetness, they focused the analyses
on the spatial variability in two different directions, one wetter and one drier, and they estimated
the annual net ecosystem exchange to -91 g C m$^{-2}$ and -62 g C m$^{-2}$ for the respective areas.
The Arctic ecosystems constitute also a source of $CH_4$ to the atmosphere even if it is not a very
large one. Saunois et al. (2020) estimated that the Northern high latitude region (60ºN - 90ºN)
contributed 4% of global emissions and emissions from wetlands are only part of the emissions
from this region. However, in the light of the vulnerability of the high Arctic permafrost areas
and considering the large carbon pool and the predicted changes in climate, a quantification and
understanding of $CH_4$ exchanges in these areas are still important. Christensen et al. (2004)
showed one example of a dramatic impact of the climate warming on the $CH_4$ emissions in a
permafrost mire in sub-arctic Sweden. The warming which is visible in this area since decades
and its impact on permafrost and vegetation changes was estimated to have caused an increase of
landscape $CH_4$ emissions in the range 22-66% in the period 1970 to 2000.
Mastepanov et al. (2008) were the first to show the importance of emissions also outside of the
growing season. They observed a large burst of $CH_4$ from a fen area in Zackenberg, Greenland
after the growing season and during the time when the soil started to freeze. This finding was
confirmed in a later paper (Mastepanov et al., 2013) and the process was hypothetically
attributed to the subsurface $CH_4$ pool. Hydrology and vegetation composition play an important
role for $CH_4$ emission and dynamics. McGuire et al. (2012) made a comprehensive summary of



$CH_4$ exchanges of the Arctic tundra showing the difference between wet and dry ecosystems; the
wet tundra emitted 5.4 to 13.0 g $CH_4$-C $m^{-2}$ during summer and 8.5 to 20.2 g $CH_4$-C $m^{-2}$
annually. The corresponding values for the dry/mesic tundra were 0.3 to 1.4 g $CH_4$-C $m^{-2}$ and 0.3
to 4.3 g $CH_4$-C $m^{-2}$, respectively. Bao et al. (2021) utilized year-around measurements of $CH_4$
fluxes from three sites of the Ameriflux network in Northern Alaska to demonstrate the
importance of the spring and autumn seasons for the annual emission. The shoulder seasons
contributed about 25% of the annual emissions and the autumn season had about three times
higher emission than the spring season. These findings increasingly emphasise the importance of
year-around measurements to fully understand the $CH_4$ controls and dynamics.

The main aim of this study is to provide another piece of the puzzle concerning $CO_2$ and $CH_4$
exchanges from different but widespread ecosystem types in the high Arctic. We hypothesise
that this moist tundra ecosystem is a net annual carbon sink and that the summer emissions of
methane will be at average levels. We made flux measurements of $CO_2$ and $CH_4$ in an moist
moss tundra ecosystem situated at Kapp Linne on the west coast of the Svalbard archipelago in
2015 and with an additional campaign in 2016. The measurements in 2015 were done using both
eddy covariance system ($CO_2$) and chambers ($CO_2$ and $CH_4$) but only chambers in 2016. We
quantify ecosystem respiration ($R_{eco}$), gross primary productivity (GPP) and net ecosystem
exchange (NEE) during the growing season based on measurements and we extend the time
period to a full year by modelling. The $CH_4$ emission was only quantified for the summer season.
We also analyze the environmental controls of the fluxes.
**2 Materials and Methods**
2.1 Research site and measurements

This study was performed in the Svalbard archipelago near the weather station Isfjord Radio
(78°03′08″ N 13°36′04″ E, alt. 7 m) which is located right on the foreland of Kapp Linné on the
island of Spitzbergen (Fig. S1). The tundra area where the measurements were performed is
located about 1 km southeast of the station. The study area consists of moist moss tundra, a
widespread ecosysetem in Svalbard (Vanderpuye et al., 2002; Ravolainin et al., 2020). The
vegetation is characterised by the moss species *Tomentypnum nitens, Sanionia uncinata* and
*Aulocomium palustre* and a sparse cover of vascular plants (20-40%), dominated by *Equisetum*
*arvense, Salix polaris* and *Bistorta vivipara*. Other vascular plant species found in the plots:
*Saxifraga cespitosa*, *Saxifraga oppositifolia*, *Silene aucaulis*, and some grass species, most likely
*Alopecurus ovatus* (previously *A. borealis*), and *Poa arctica*. The vegetation analysis was made
from photographs of chamber location plots taken between 26 June and 2 July 2015 (see Figs.
S4a-4y in Supplement).

The net ecosystem exchange of $CO_2$ was measured with an eddy covariance (EC) system located
centrally on the moss tundra (78°03′28.6″ N 13°38′40″ E). The sonic anemometer (USA-1;
Metek GmBH, Germany) was mounted on top of a tripod (see Fig. S1) at 2.7 m height. The $CO_2$
and $H_2O$ concentrations were measured with an open path sensor (LI-7500; Li-Cor Inc., USA)
placed just beneath the sonic and inclined about 30º pointing towards east. Radiation
components, incoming and outgoing short-wave and long-wave (CNR-4; Kipp & Zonen, the
Netherlands) were measured at 2.0 m height above ground with the sensor directed towards



south. All sensors were connected to a datalogger (CR-1000; Campbell Scientific, USA) which
was powered by a solar panel and a battery. The EC sensors were sampled and stored at 10 Hz
and all oter sensors were sampled at 0.1 Hz with storage of 30 min mean values. These
measurements were made from 25 June to 17 September 2015.

The soil efflux of $CO_2$ and $CH_4$ was measured with a dark chamber connected to a gas analyzer
(Ultraportable Greenhouse Gas Analyzer; Los Gatos Research, USA) on 24 locations within the
EC average footprint area. A circular thin-steel frame, 15 cm in diameter and 15 cm high, was
inserted ca 5 cm into the ground in each location. The sharp edge of the frames made it easy to
insert them into the ground without damaging the vegetation and with minimal soil disturbance.
A picture was taken of each frame (see Supplement) for documentation of vegetation and for
calculation of different indexes. The chamber was also made from steel and it had a rubber seal
in the end facing the frame (Fig. S2) to make it air tight when mounted on the frame. The volume
of the chamber and the part of the frame raised above the surface was 5.3 L. A small fan was
installed inside the chamber to provide good mixing of the air during measurement. A small
weight (stone) was placed on top of the chamber during measurement to prevent it from moving
due to wind gusts. During concentration measurement air was circulated in a closed loop
between the chamber and the gas analyzer in ca. 10 m long 4 mm diameter polyethene tubes (see
Fig. S2). The air flow through the analyzer was ca 1.2 L min$^{-1}$. The chamber was ventilated in
the free air about 1 minute before each measurement which lasted for 5 minutes. The
concentrations were recorded and stored once per second by the gas analyzer. The time stamp of
the recorded data was used to identify measurement cycles for analysis of fluxes.

The chamber measurement positions were selected in the following way. The frames were
grouped in two sections, one north-east and one south-west of the flux tower since it was
expected that the main wind direction would be along that direction. Each group was then split
into three subsections with four measurement points within each one of them. The locations were
named S1:1-S1:4, S2:1-S2:4, S3:1-S3:4, N1:1-N1:4, N2:1-N2:4 and N3:1-SN3:4. The four
measurement points within each subsection were then placed along a transect with 3-4 m
between each point. This way it was possible to measure all four chamber locations without
having to move the whole measurement system. Chamber measurements were made in three
separate campaigns: mid-summer (26 June to 2 July 2015), late-summer (25-27 August 2015)
and early-summer (14-15 June 2016). Each location was measured three times during each one
of the three campaigns, a total of 216 measurements. Besides gas concentrations, also soil
temperature (5 cm), soil moisture (0-5 cm) and active layer depth was measured during each
campaign.

Meteorological data needed for analyses and gap-filling were obtained as follows: Hourly air
temperature and relative humidity from Isfjord radio, half-hourly global radiation from
Adventdalen, daily snow depth and ground ice conditions from Svalbard airport and monthly
precipitation from Isfjord radio and Barentsburg. The distance between the measurement site and
these stations are; Isfjord radio, 1 km, Barentsburg, 13 km, Svalbard airport, 46 km and
Adventdalen, 50 km. Data sources are given in Acknowledgement.

**3. Data analysis**



The rawdata from the eddy covariance flux measurements were analysed using the Eddypro
software version 6.1.0 (Li-Cor, 2016). Correction was made for the impact of the additional heat
flux in the sensor path of the open path analyzer on the flux calculations according Burba et al.
(2008). Gap filling during the measurement period was made using the REddyProc online eddy
covariance data processing tool developed at the Max Planck Institute for Biogeochemistry
(Wutzler et al., 2018) without u* correction since we could not identify any threshold for u*. The
u* threshold is generally low for low and smoth vegetation (Pastorello et al., 2020) and for a
wind exposed site as ours, it is not surprising that such threshold could not be found. Only data
of highest quality, i.e. class=0 was retained for the gap filling and further analyses. Gap filling
outside of the EC measurement period to obtain the carbon balance for a full year was made
using empirical relationships for $R_{eco}$ and GPP (see below).

For flux footprint calculations the roughness length ($z_0$) is needed and it was calculated from the
wind profile relationship in near neutral (-0.01<z/L<0.01) conditions:

$$z_0 = \frac{z_m}{e^{\left(u(z) \cdot \frac{k}{u^*}\right)}} \qquad\qquad (1)$$

where $z_m$ is measurement height, u(z) is wind speed at height z, k is von Karman's constant and
u* is friction velocity. We used the flux footprint prediction (FFP) online tool by Kjun et al.
(2015) to calculate the footprint climatology.

The fluxes from the chamber measurements were estimated from the time change of the
concentrations using linear regression. Every individual measurement was inspected and
evaluated manually. These inspections showed that 50 seconds for $CO_2$ and 100 seconds for $CH_4$
were optimal to obtain near perfectly linear responses a few seconds after the chamber had been
placed on the frame. The slopes of the regressions were then used to calculate fluxes per unit
surface area. The flux detection limits for $CO_2$ and $CH_4$ were calculated in the following way:
first the peak-to peak variation in the respective gases were determined when the chamber was
ventilated in the free air and when conditions were steady. Then 20 sets of artificial 'fluxes' for
each gas species were estimated based on 100 randomly generated concentrations for each data
set. The peak-to-peak difference was used as seed (input) for the randomly generated values. The
95% value of the distribution of these randomly generated fluxes was taken as the flux detection
limit for the respective gas.

The pictures of the vegetation inside of the chamber frames were analysed using the ImageJ
(https://imagej.net) public domain software. The camera color channel information (digital
numbers for Red (R), Green (G) and Blue (B) channels) was collected from the JPEG pictures.
This type of pictures is for instance used in studies that are tracking the phenological
development of vegetation (e.g. Richardson et al., 2009). The so-called green index (GI) is
applied to detect differences in greenness of vegetation:

$$GI = G/(R+G+B) \qquad\qquad (2)$$



This index was also estimated for the central footprint area (100 m radius) of the flux
measurement location using a picture taken at 160 m above the altitude of the measurement area.
Forward stepwise linear regression (Sigmaplot 12.5) was used to analyze the dependency of the
$CO_2$ and $CH_4$ fluxes on environmental variables. We tested for air temperature ($T_a$), soil moisture
($\theta$), soil temperature ($T_s$), active layer depth (ALD), measurement location ($S_{id}$) and GI.
For gap filling of $R_{eco}$ we only had access to air temperature with full annual coverage and, thus,
we could only use this driver for estimation of the $R_{eco}$. The measured chamber $CO_2$ fluxes were
fitted to the Lloyd & Taylor (1994) model with air temperature ($T_a$) as independent variable:

$$FCO_2 = a \cdot e^{b(\frac{1}{56.02} - \frac{1}{T_a + 46.02})} \tag{3}$$
During the EC measurement period (25 June to 17 September 2015) the GPP was estimated as:

$$GPP = NEE_f - R_{eco} \tag{4}$$
Where $NEE_f$ is the gap filled NEE according to Wutzler et al. (2018). This way $R_{eco}$ and GPP
become consistent with the measured and gap filled NEE. For the time before and after this
period NEE was estimated as the sum of modelled $R_{eco}$ and modelled GPP.  The data for the GPP
model was derived from:

$$GPP_m = NEE_m - R_{eco} \tag{5}$$
Where $NEE_m$ is the measured net ecosystem exchange. The $GPP_m$ was then fitted to a light
response function:

$$GPP_m = c1 + c2 \cdot c3/(c2 + R_g) \tag{6}$$

**4 Results**
For $CO_2$ exchanges and partitioning we combined the soil efflux measurements with the chamber
system with the eddy covariance flux measurements. This was crucial for the partitioning and for
gap filling because from 20 April to 20 August at this location the sun is above the horizon 24
hours of the day and this means that there were few occasions of dark nighttime measurements
with the eddy covariance system and all of these were collected at the very end of the summer.
We consider the chamber measurements that were distributed across the summer to be more
representative of $R_{eco}$ for this location.
For $CH_4$ exchanges we don't have any eddy covariance measurements so we present only
chamber data for this variable.
4.1Weather
The mean annual temperature at Kapp Linne was -1.5 ℃ during 2015 which was 3.5 ℃ higher
than the long-term mean (1961-1990) of -5.1 ℃. The summer (June-August) mean of 5.5 ℃ was



2.0 ºC higher than the long-term mean for the same time period (Fig. 1). The summer
precipitation in 2015 was much lower, 58 mm as compared to the long-term precipitation which
was 121 mm. The annual precipitation was also lower, 431 mm compared to the long-term
precipitation which was 514 mm.

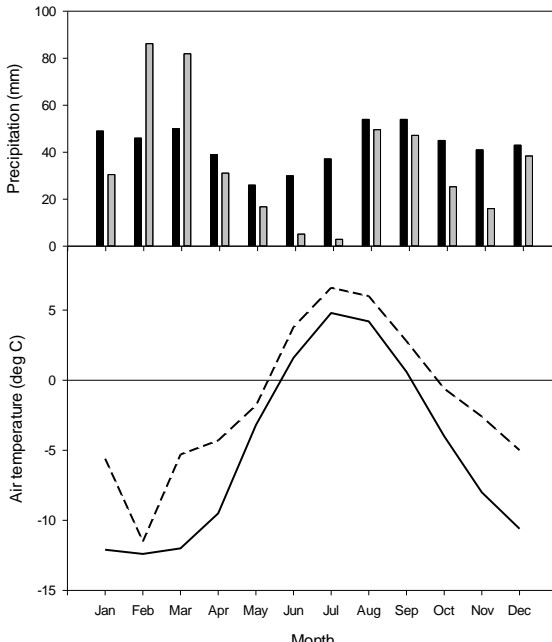

Figure 1. Monthly precipitation (top): Long-term average 1961-1990 black bars and 2015 grey
bars. Data from Barentsburg for January-May, from Isfjord Radio for June-December. Mean
monthly air temperature (bottom): Solid line is long-term average 1961-1990 and dotted line is
2015. Data from Isfjord Radio which is located about 1 km west of the investigation area.
We defined the start of the growing season (the period during which vegetation is
photosynthesizing) in two different ways. The first (denoted Season 1, day no. 140; see Fig. 2)
based on when daily air temperature started to stay above zero more steadily and the second
(denoted Season 2, day no. 160) when most of the snow had disappeared. The ending of the
growing season was defined as when the air temperature fell more steadily below zero, when
ground ice began to establish and when a significant snow pack was established (day no. 284;
Fig. 2).



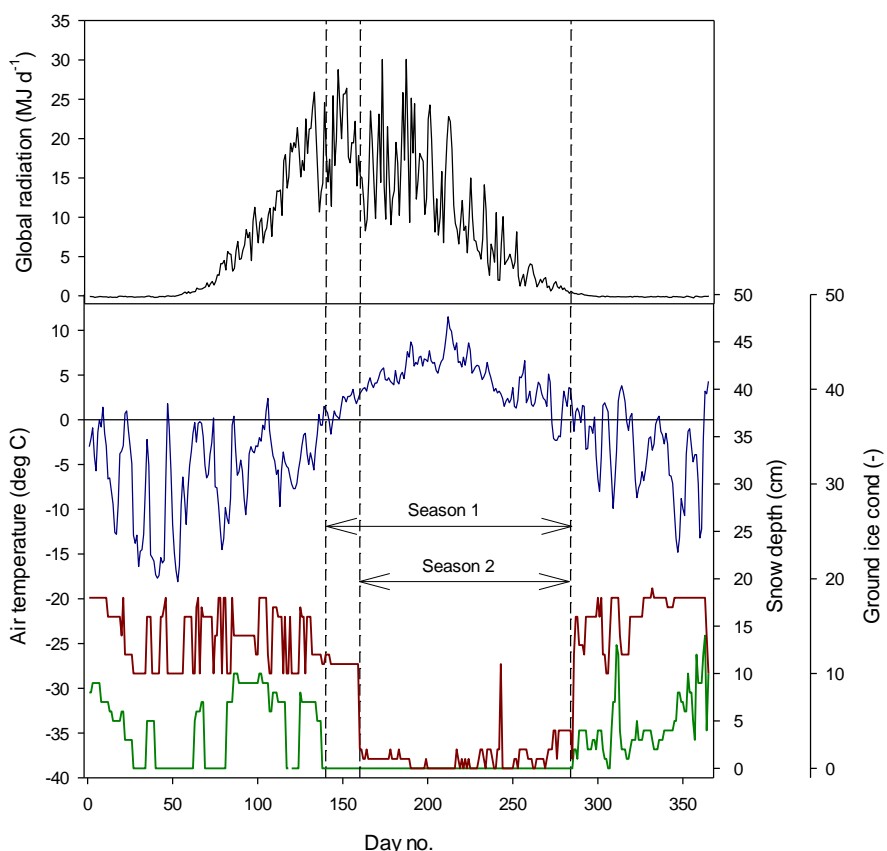

Fig. 2 Weather conditions during 2015. Top panel: Mean daily global radiation at Adventdalen.
Bottom panel: Mean daily air temperature at Isfjord Radio (blue), snow depth (red) and ground
ice conditions (green) at Svalbard airport close to Longyearbyen. The ground ice condition is
scaled from 0 to 20 where 0 is no snow or ice on the ground and 20 indicate a complete cover of
snow or ice.
4.2 Flux footprint and greenness
The footprint climatology shows a good representativity of the moss tundra surface by the EC
measurements with 60-70% of fluxes emanating from areas well within the border of the tundra
(Fig. 3). The mean green index for a circular area with radius of 100 m centered at the flux tower
was 0.34 which corresponded exactly to the mean value for all chamber locations. The GI for the
24 chamber locations varied between 0.316 and 0.369. We observed a good (visual) correlation
between GI and coverage of green plants (see Figures S4a-S4y of chamber location pictures and
GI).





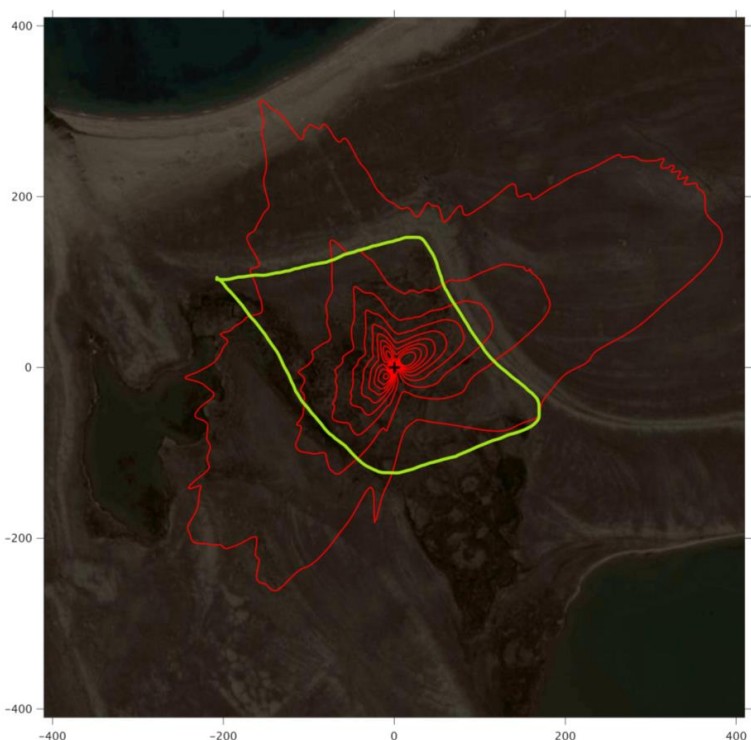

Figure 3. The footprint climatology with red contour lines 10-90%. The area within the green
line mark the heart of the moss tundra. The scale (m) is shown on the outer borders of the
picture.
4.3 $CO_2$ exchanges
The $CO_2$ fluxes from the chamber measurements showed quite large variation over time (Fig. 4)
and across sampling locations (Fig. 5). The mean $CO_2$ flux of all samples was 0.81±0.11 µmol
$m^{-2}s^{-1}$. The uncertainty is given as the 95 confidence limit.

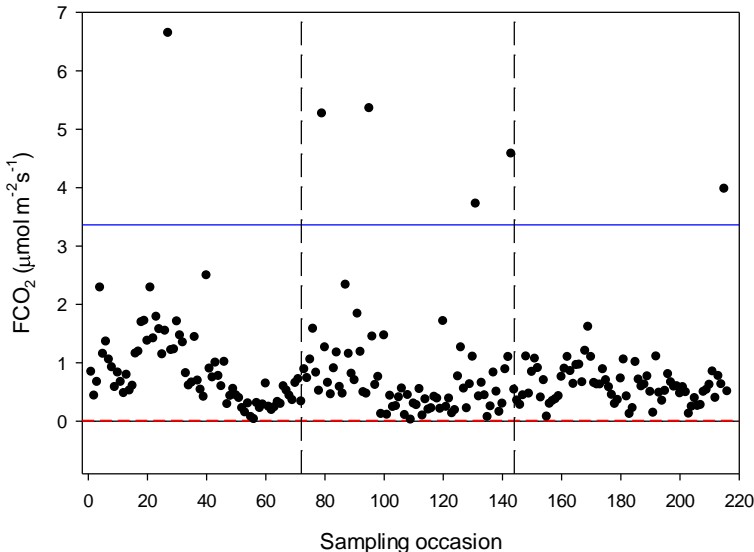

Figure 4. Measured $CO_2$ exchange from the 24 sampling points using dark chamber and portable
gas analyzer. The dashed red line indicates $CO_2$ flux detection limit and the blue line represents
3xS.D. of all data points. The dashed vertical lines separate sampling periods from left to right:
early summer, mid-summer and late-summer.

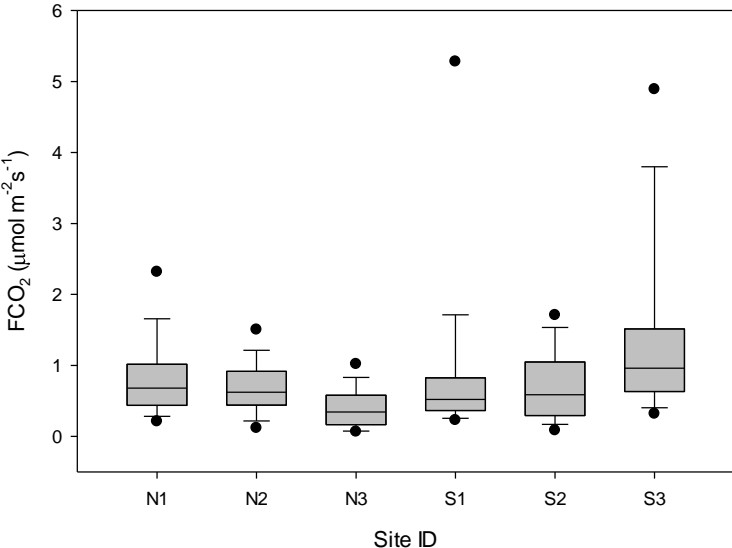

Figure 5. Box plot of $CO_2$ fluxes per sampling location named N1-N3, S1-S3. The boundaries of
the grey boxes represent the 25% and 75% percentiles, the line represent the median, whiskers
above and below the boxes indicate the 10% and 90% percentiles. Outlaying points are also
shown.






Of the tested environmental variables $T_a$, $\theta$, $T_s$, ALD, $S_{id}$ and GI it was only $T_a$, $\theta$ and GI that
contributed positively and significantly in decreasing order to explain the variability of the $CO_2$
flux (Table 1).

Table 1. Result of stepwise linear regression with $CO_2$ flux as dependent variable. Normality test
failed but significance in all variables was confirmed with Wilcoxon Signed rank tests.

| Variable | Partial-$R^2$ | Probability (p) |
|----------|---------------|-----------------|
| $T_a$ | 0.190 | <0.001 |
| $\theta$ | 0.037 | 0.002 |
| GI | 0.023 | 0.002 |


Ideally all of these variables should be used in a model to estimate $R_{eco}$ for gap filling purposes
but we could only use air temperature since this was the only variable that we had access to with
complete coverage for a full year. The Lloyd &Taylor model (Eq. 3 & Fig. 6)) was thus used to
estimate ecosystem respiration for 2015 using half-hourly air temperature as input.

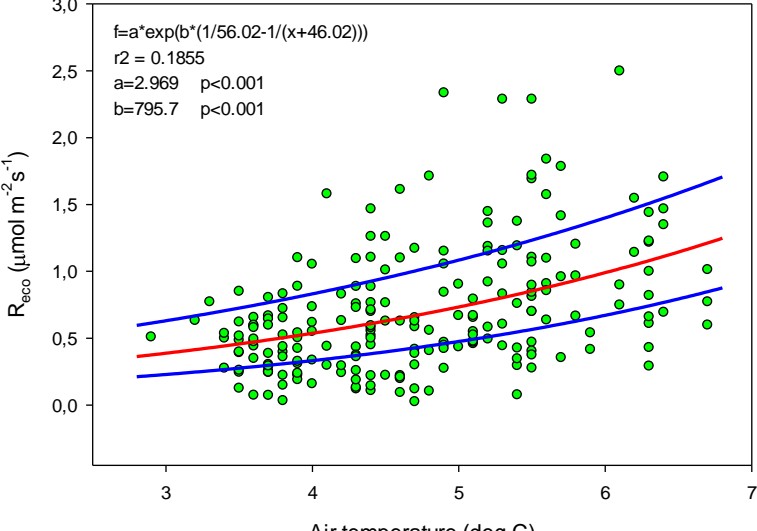

Figure 6. Measured ecosystem respiration (green dots) plotted against air temperature. The red
curve is the fitted equation and the blue curves are the corresponding boundaries when
considering the standard deviation of the parameters.

The modelled gross primary productivity (Eq. 6; $GPP_m$) had a small offset when global radiation
was zero (Fig. 7). This offset was adjusted for when the model was applied for gapfilling so that
GPP become zero during nighttime.



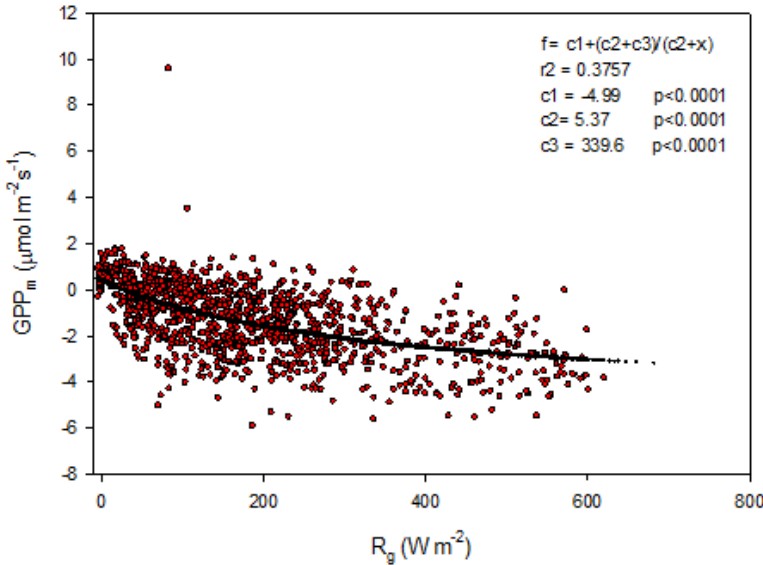

Figure 7. $GPP_m$ plotted against global radiation; red symbols are estimated values according to
eq. (5) and the black symbols are the fitted model.
We assumed that GPP was zero for the periods outside of the growing season and that our $R_{eco}$
model was valid during winter as well as during growing season. However, during winter the air
temperature is not the appropriate driver for respiration because of the insulating effect of the
snow cover. We did not have access to soil temperature from our site but data from Adventdalen
which is located about 60 km east from our site, showed that the soil surface temperature was
close to zero degrees below the snow pack during the winter (N. Pirk, unpublished data). Thus,
we assumed that the situation would be similar in our site and used zero degrees as driver for
respiration.
The mean bi-weekly fluxes show that NEE is negative from about one week into June until one
week into August (Fig. 8). The mean NEE is relatively constant during this period with a low -
0.5 $\mu mol\ m^{-2}s^{-1}$. The maximum bi-weekly GPP is about -2.5 $\mu mol\ m^{-2}s^{-1}$ while the corresponding
$R_{eco}$ is about 2.0 $\mu mol\ m^{-2}s^{-1}$. The GPP become positive during one period in the autumn
indicating an underestimation of $R_{eco}$ during that time.

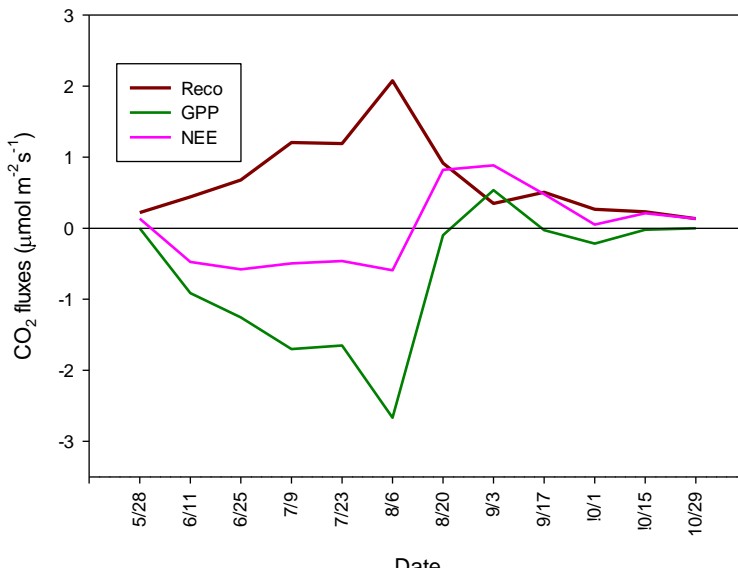

Figure 8. Bi-weekly gap filled $CO_2$ fluxes for season 2 (see Fig. 2) at Tunsjömyren, Kapp Linne
during 2015.
The annual modelled and gap filled NEE was negative, -12.4 g C m$^{-2}$ for season 1 and positive,
24.7 g C m$^{-2}$ for season 2. The gapfilled NEE (Table 2) during the summer (June-August) was -
37 g C m$^{-2}$ or -0.40 g C m$^{-2}$ day$^{-1}$ which is good agreement with the measured NEE (25 June -31
August) with a mean daily uptake of -0.40 g C m$^{-2}$ day$^{-1}$. A summary of all components for the
different seasons are presented in Table 2.
Table 2. Summary of annual and seasonal C-fluxes from Kapp Linne.

| Period | Component (g C m$^{-2}$) | Season | |
|---|---|---|---|
| | | 1 | 2 |
| Winter | Reco | 29.5 | 31.5 |
| | GPP | 0 | 0 |
| | NEE | 29.5 | 31.5 |
| Growing season | Reco | 114.9 | 109.5 |
| | GPP | -156.8 | -116.3 |
| | NEE | -41.9 | -6.8 |
| Summer (June-August) | Reco | 97.8 | 97.8 |
| | GPP | -134.8 | -134.8 |
| | NEE | -37.0 | -37.0 |
| Annual | Reco | 144.4 | 141.0 |
| | GPP | -156.8 | -116.3 |
| | NEE | -12.4 | 24.7 |






4.4 Temperature sensitivity of $R_{eco}$ and GPP


The temperature sensitivity of the $R_{eco}$ is already given by the fitted Lloyd & Taylor (1994)
equation. In the absence of long time series of measurements during multiple year were natural
climate variability could be used to assess temperature sensitivity of GPP we approached this
problem in the following way. We normalize GPP for its dependence on radiation by estimating
the difference between the 'measured' GPP and the model which only depends on radiation (see
Fig. 7). The resulting normalized GPP show a dependence on air temperature (Fig. 9) with values
becoming more negative with increasing temperature. We fitted the same type of model to these
data as for the $R_{eco}$ to be able to compare sensitivities to temperature.

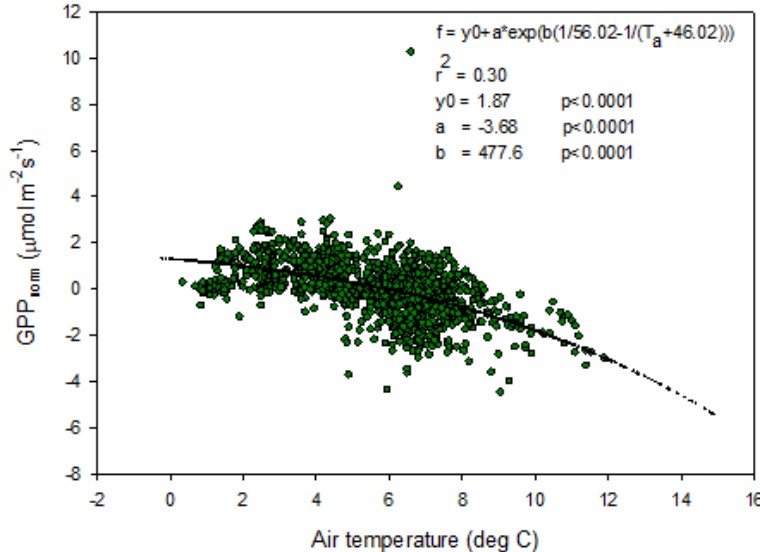

Figure 9. Normalized GPP plotted against air temperature and with the fitted exponential model.



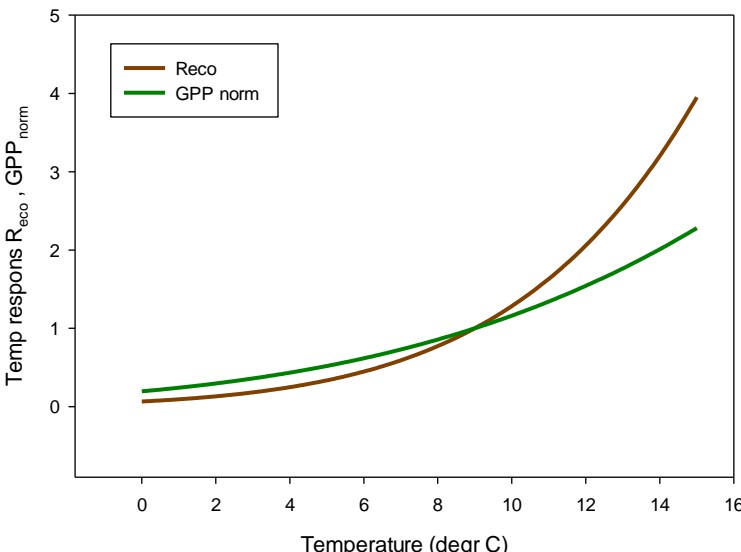

Figure 10. Temperature sensitivity for $R_{eco}$ (brown) and $R_g$-normalized (positive) GPP (green).
In Fig. 10 we reversed the sign of the GPP temperature response function to make it more easily
comparable with the $R_{eco}$ response model. The temperature sensitivity ($\mu$mol m$^{-2}$s$^{-1}$ K$^{-1}$) can be
estimated from the slope of these curves and the sensitivity is slightly higher for GPP than for
$R_{eco}$ in the interval 0 – 4.5 ºC, thereafter the difference is small up to about 7 ºC then it began to
raise rapidly for $R_{eco}$. We tested what impact this could have by increasing the measured half-
hourly air temperature by 1 ºC and found that during the growing season (season 2) the GPP
increased by -3.89 g C m$^{-2}$ and $R_{eco}$ by 3.53 g C m$^{-2}$. Thus, a minor increase of GPP compared to
$R_{eco}$. However, a one-degree higher winter temperature resulted in an addition respiration of 9 g
C m$^{-2}$. Thus, an estimated loss of 8.5 g C m$^{-2}$ for the whole year.
4.5 CH$_4$ exchanges
The CH$_4$ fluxes from the chamber measurements showed large variation over time (Fig. 11) and
across sampling locations (Fig. 12). The mean CH$_4$ flux of all samples was 0.00051±0.00024
$\mu$mol m$^{-2}$s$^{-1}$. The uncertainty is given as the 95% confidence limit. Setting all fluxes that fell
within the flux detection limits to zero changed the mean value with -0.2%. Assuming that the
mean flux was representative for the whole of growing season 1, the total CH$_4$ summer emission
was 0.039 to 0.164 g CH$_4$ m$^{-2}$. Converting this to CO$_2$ equivalents (CO$_2$-eq) we get a range of 1.1
to 4.6 g CO$_2$-eq for the summer and if we add also a possible winter emission of 22% of the
annual (following Bao et al., 2021) we obtain an annual mean of 3.2±2.0 g CO$_2$-eq using a
GWP$_{100}$ of 27.9 (Arias et al., 2021). The corresponding value using GWP$_{20}$ of 81.2 (Arias et al.,
2021) is 9.3±5.8 g CO$_2$-eq for the annual emission.
We also noticed a clear trend during the summer with highest fluxes in mid-June and then
decreasing during the following two sampling occasions. The respective mean values with 95%



confidence intervals for the three sampling periods were $0.00121\pm0.000512$ µmol m$^{-2}$s$^{-1}$(June 14-
15), $0.000332\pm0.000465$ µmol m$^{-2}$s$^{-1}$ (June 26- July 2) and $-0.00000781\pm0.0000936$ µmol m$^{-2}$s$^{-1}$(August 25-26).
For CH$_4$ exchanges we found *ALD, $\theta$* and *GI* to contribute significantly to explain the variance of
the flux (Table 3). The CH$_4$ flux responded negatively to increasing ALD and positively to $\theta$ and
*GI*.
Table 3. Result of stepwise multiple linear regression with CH$_4$ flux as dependent variable.
Normality test failed but significance in all variables was confirmed with Wilcoxon Signed rank
tests.

| Variable | Delta-$R^2$ | Probability (p) |
|---|---|---|
| ALD | 0.175 | <0.001 |
| $\theta$ | 0.025 | 0.01 |
| GI | 0.020 | 0.004 |


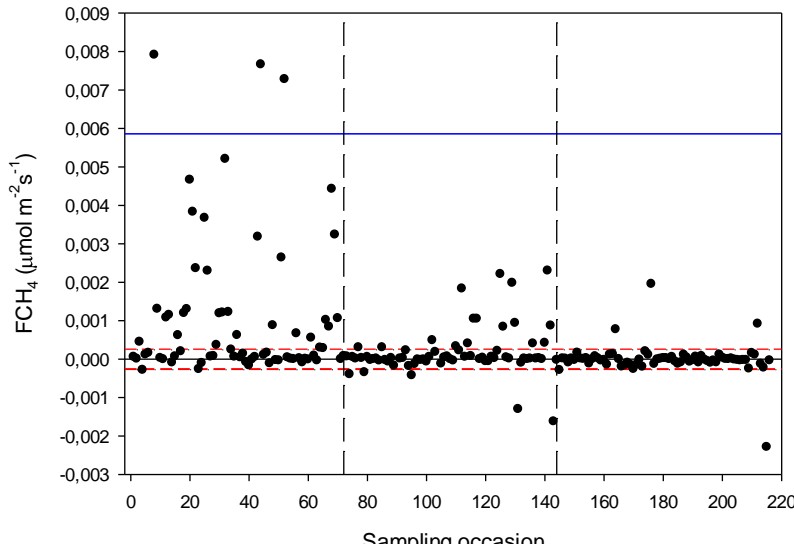

Figure 11. Measured CH$_4$ exchange from the 24 sampling points using dark chamber and
portable gas analyzer. The dashed red lines indicate CH$_4$ flux detection limit, (i.e. inside the
limits of detection the exact numbers are highly uncertain) and the blue line represents 3xS.D.
The dashed vertical lines – same as in Fig. 4.


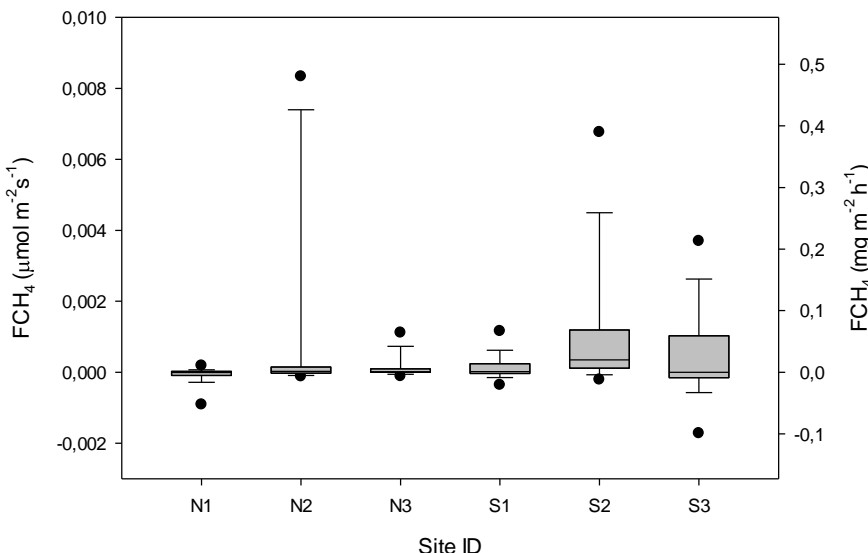


Figure 12. Box plot of $CH_4$ fluxes per sampling location named N1-N3, S1-S3. The statistics
includes also the data that fall within the flux detection limits. The boundaries of the grey boxes
represent the 25% and 75% percentiles, the line represent the median, whiskers above and below
the boxes indicate the 10% and 90% percentiles. Outlaying points are also shown.

**5 Discussion**

    5.1 Annual and seasonal $CO_2$ fluxes

We focus our discussion mainly on comparison with other tundra sites located in the North
Atlantic area since these sites are influenced by the North Atlantic Current with its impact on
weather patterns and climate. This limits the comparisons to sites in Greenland, Svalbard and
Northern Scandinavia. However, we broaden the comparison a bit by adding two sites from
Alaska.

Our annual NEE was in the range -12.4 to 24.7 g C m$^{-2}$ depending on definition of growing
season (Table 2). We judge the latter value to be more realistic since season 1 includes an
unrealistically high GPP when there is still a snow cover on the ground in early spring. We used
the high and low estimates of the fitted functions for R$_{eco}$ (see Fig. 6) to assess the sensitivity of
annual NEE to uncertainty in winter respiration and we found the range to be between 17.1 g C
m$^{-2}$ to 32.9 g C m$^{-2}$.

Lund et al. (2012) found that the start of the uptake period was strongly correlated with start of
the snowmelt for the fen in Zackenberg, NE Greenland. They defined the start of snowmelt as
the day when snow depth was <0.1 m. This coincides very well with our definition of start of
growing season 2 (see Fig. 2). Soegaard and Nordtroem (1999) reported an annual NEE of -64.4
g C m$^{-2}$ for the fen in Zackenberg and Pirk et al. (2017) reported -82 g C m$^{-2}$ for an alluvial fen in





Adventdalen, Svalbard, not far from Kapp Linne. For a site on the west coast of Greenland,
Disco island with heath vegetation, Zhang et al. (2019) reported an annual NEE of -25±15 g C
$m^{-2}$. Christensen et al. (2012) reported a range of -20 to -95 g C $m^{-2}$ for annual NEE in a palsa
mire in Abisko, Northern Sweden. Our results are closer to the values found for a sparsely
vegetated catchment area in Bayelva, Ny-Ålesund were Lüers et al. (2014) reported annual NEE
to be 0 g C $m^{-2}$. If we go beyond the North Atlantic area to the low Arctic region in North
America we can find sites that has a positive NEE on annual basis, 13.6 g C $m^{-2}$ for a tussock
tundra near Atquasuk, Alaska (Oechel et al., 2013) and 21-61 g C $m^{-2}$ for a heath and 2-82 g C
$m^{-2}$ for a wet sedge ecosystem in Imnavait creek (Eurkirchen et al., 2012).
Lund et al. (2012) analysed 10 years of EC flux measurements from a heathland in Zackenberg
and they reported a NEE range of -39.7 to -4.3 g C $m^{-2}$ for the growing season. Our result for the
growing season NEE of -6.8 g C $m^{-2}$ (Season 2; Table 2) fall within the same range but it was
only two years out of ten that showed that low uptake in Zackenberg heath. Their measured
growing season GPP was in the range of -95.4 to -54.1 g C $m^{-2}$ and the $R_{eco}$ was in the range of
37.7 to 63.8 g C $m^{-2}$. Our corresponding values were -116.3 g C $m^{-2}$ for GPP and 109.5 g C $m^{-2}$
for $R_{eco}$. López-Blanco et al. (2017) presented data over a period of eight years of EC flux
measurements from Kobbefjord, SW Greenland over an area of mixed fen and heath vegetation.
Their growing season ranges were; for NEE -74.2 to -45.9 g C $m^{-2}$, for GPP -316.2 to – 181.8 g
C $m^{-2}$ and for $R_{eco}$ it was 144.2 to 279.2 g C $m^{-2}$ excluding 2011 which was anomalous because
of a pest outbreak and 2014 which did not have a full growing season.
Our EC measurements of summer (June-August) NEE of -37 g C $m^{-2}$ (Table 2) is in-between
ranges reported for fen type of vegetation in NE Greenland; -96.3 g C $m^{-2}$ (Soegaard and
Nordstroem 1999) to -50 g C $m^{-2}$ (Rennermalm et al. 2005) and heath vegetation; -1.4 to -18.9 g
C $m^{-2}$ (Groendahl et al. 2007).
It is difficult to compare growing season values because they are rarely defined the same way.
Only small differences in definition of start and end of growing season can have a large impact
on the NEE values since NEE is the sum of two large components of almost equal size and of
different sign. In our case a 20 days difference in the beginning of the season changes growing
season NEE from -12.4 to 24.7 g C $m^{-2}$. It is also difficult to compare GPP and $R_{eco}$ for any
season since the methods to split NEE into components differ from case to case. The most
reliable comparison is probably for summer season (June – August) since most studies represents
this period best in terms of measurement coverage and quality. So, with this in mind we are
pretty confident with placing the C-exchange rates of the moss tundra intermediate between fen
and heath type of vegetation in the North Atlantic region.

5.2 $CH_4$ fluxes

Our estimated growing season $CH_4$ flux of 0.08 g C $m^{-2}$ is very low compared to most other
methane emitting tundra sites; the Zackenberg fen site emitted $CH_4$ in the range 1.4 to 4.9 g C $m^{-2}$
(Mastepanov et al. (2013), Jackowicz-Korczynski et al. (2010) reported 20.1 to 25.1 g $CH_4$ $m^{-2}$
for the Stordalen mire in Northern Sweden.  For three different sites in northern Alaska, Bao et
al. (2021) reported annual emissions between 1.8 and 8.5 g $CH_4$ $m^{-2}$ which corresponds to 0.94
and 4.5 g $CH_4$ $m^{-2}$ for the growing season based on their estimate that growing season emissions



are 52.6% of the annual emissions. Sachs et al. (2008) measured $CH_4$ exchanges with EC method
in a northern Siberian polygon tundra and found generally low fluxes of about 18.5 mg $CH_4$ m$^{-2}$
day$^{-1}$ with little variation over the growing season. This rate adds up to 2.3 g $CH_4$ m$^{-2}$ for their
four months long growing season.
It should be pointed out that we did not perform measurements during the shoulder seasons
meaning that we probably underestimate the seasonal total. Importance of shoulder seasons was
first pointed out by Mastepanov et al. (2008) which discovered a large burst of $CH_4$ at and after
the onset of soil freezing. One interesting observation is that the main part of our $CH_4$ flux
occurred during the sampling period 14-15 June 2016 which is about 30 days after snow melt.
This is the time of the season when $CH_4$ emissions normally are peaking (Mastepanov et al.
2013). After that, the rates dropped to practically zero in late August (see Fig. 11).
If we sum up the annual net $CO_2$ and $CH_4$ fluxes expressed as $CO_2$-eq we find that the moss
tundra is emitting in total 60 g $CO_2$-eq of which the methane stands for 7%. So even if the $CH_4$
fluxes are small, it still represents a significant global warming impact in relative terms.
The comparison between the different sites are hampered by the fact that they in most cases
belong to different bioclimatic subzones with differences in climate and vegetation (Walker et
al., 2005). The only site besides Kapp Linne that belong to subzone B is the one in Ny Ålesund.
The other high Arctic sites Adventdalen and Zackenberg both belong to subzone C, the
intermediate high/low Arctic sites Kobbefjord and Disco Island belongs to subzone D
respectively C/D. The low Arctic site Atqasuk belong to subzone D and the Imnavait Creek
belong to subzone E. The sub-Arctic Abisko is not classified by Walker et al. (2005) but based
mean July air temperature it should belong to subzone E. These differences in climate and
vegetation should be kept in mind when comparing results from different sites.
5.3 Environmental controls of fluxes
A key issue in high Arctic is how ecosystems with soil that contain large amounts of frozen
carbon will respond to warming. A recent report about the future climate of Svalbard (Hanssen-
Bauer et al. 2019) show that appalling changes are at risk to occur. By 2071-2100 compared to
1971-2000 the mean annual temperature is estimated to increase by 7 °C to 10 °C for the medium
and high emission scenarios, respectively. Precipitation is also estimated to increase by 45%
respectively 65% for these scenarios. Such large changes will of course also have a lot of other
impacts as well for instance shorter snow season, more erosion and sediment transport, changes
in vegetation composition and growth etc etc. Assessment of such large changes are very
difficult and is far beyond the scope of this paper. We have however shown that for a smaller
temperature increase of 1 degree, the impact on the net carbon balance during the growing
season will be minute; the increase in ecosystem respiration is compensated for by a
corresponding, or actually slightly larger increase of gross primary productivity. Similar
compensation effect was obtained for a heath site in Zackenberg by Lund et al. (2012). They
used multi-year measurements to assess the effect of changes in temperature on the growing
season fluxes. But, if we also consider an increase in temperature during winter, it is most likely
that the annual NEE becomes weakened. It is not unlikely that the impact of climate change with





higher temperature that is already a reality in Svalbard can be the reason why the annual NEE
now is positive, i.e. the moss tundra is a GHG source of $CO_2$ to the atmosphere.
We found that air temperature was the main control of ecosystem respiration followed by soil
moisture and greenness index (Table 1). We had expected that soil temperature should contribute
significantly to explain the variations in $R_{eco}$ but it did not. Cannone et al. (2019) showed that
ground surface temperature at 2 cm depth contributed significantly to explain $R_{eco}$ in nearby
Adventdalen during early, peak and late parts of the growing season. In their study soil moisture
was also significant during peak and late seasons. One possible explanation to this difference in
responses could be that our soil temperature was measured at 5 cm depth and that air temperature
was more representative for the microbial processes taking place in or near the soil surface.
Interestingly, GI contributed significantly to explain variations in $R_{eco}$. The GI was clearly
correlated with the abundance of Salix polaris (see Supplement) and thus we interpret the
positive correlation between GI and $R_{eco}$ to be an effect of increasing contribution by autotrophic
respiration to the total respiration.
We found no significant correlation between $CH_4$ emission and temperature. The best
explanation was by active layer depth followed by soil moisture and GI (Table 3). But it should
be pointed out that ALD and θ are not independent from each other and that ALD can be
regarded as a proxy for any seasonal variability, like plant phenology. Soil moisture decreases
with increasing active layer depth. The correlation between GI and $CH_4$ emission is probably
also connected with abundance of Salix polaris which is a vascular plant. Vascular plants are
since long mentioned as a pathway for $CH_4$ from the soil interior to the atmosphere in wet tundra
ecosystems (e.g. Schimel, 1995) but it could also be an effect of mediation of soil by the root
exudation of organic acids as mentioned by Ström et al. (2012). However, we have not found any
studies supporting the latter hypothesis concerning Salix polaris.
**6 Conclusions**
Our analyses of EC and chamber flux measurements have shown that the moss tunda on Kapp
Linne is a small source of $CO_2$ and an even smaller source of $CH_4$ on an annual basis.
Concerning the magnitude of the $CO_2$ exchanges during summer we find it to be in between
those of fens and heath ecosystems located in the North Atlantic region. The $CH_4$ exchange is
much lower than for other tundra ecosystems in the region.
The temperature sensitivity for $CO_2$ exchange was slightly higher for GPP than for $R_{eco}$ in the
low temperature range of 0-4.5 ºC, almost similar up to 7 ºC and thereafter it was considerably
higher for $R_{eco}$. The consequence of this, for a small increase in air temperature of 1 degree (all
other variables assumed unchanged) was that the increases in the two fluxes practically evened
out during the growing season. But a warmer winter period would probably result in an increased
loss of carbon. We cannot rule out that the reason why the moss tundra is a net source today is an
effect of the warming that has already taken place in Svalbard.
The analysis of which environmental factors that controlled the small-scale fluxes showed that
air temperature dominated for $R_{eco}$ and active layer depth for $CH_4$ but we also found that
greenness index significantly explained part of the variation in these fluxes. For $R_{eco}$ we
attributed this to an increased share of autotrophic respiration to the total and for $CH_4$ we



hypothesized that the abundance of the woody shrub *Salix polaris* effected the exchange either through internal plant pathway for methane or through increased provision of C substrate to the anaerobic microbial community stimulating the production of methane. This finding is an indication that modeling of $CO_2$ as well as of $CH_4$ fluxes can be improved by also considering differences and changes in greenness of the vegetation.

## 7 Supplement

The supplement contains some additional photographs of equipment, site and color photograps of vegetation within the frames used for chamber measurements.

## 8 Data availability

Data can be obtained from https://zenodo.org (10.5281/zenodo.5704508).

## 9 Author contribution

AL designed the study and wrote the manuscript. NP and AL performed the EC measurements and analysed the EC data. ISJ did the vegetation characterization. AL, CS, LK and MBN performed the chamber measurements. All authors have read and commented the manuscript.

## 10 Competing interests

We declare no competing interests.

## 11 Acknowledgments

This work did not receive any other funding except salaries for the authors from their respective organizations. Observations of air temperature, relative humidity, precipitation, ground ice conditions and snow depth were obtained from Norwegian Centre for Climate Services (NCCS) and provided under licence CC BY 4.0. Global radiation data from Adventdalen was obtained from the University Centre in Svalbard (UNIS). Thanks to associated professor Jonas Åkerman, Lund University for support with information about the site.

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
