# Peer review of "CO2 and CH4 exchanges between moist moss tundra and atmosphere on Kapp Linne, Svalbard"

_Biogeosciences, 2021_

## Referee Comment (RC1)

Lindroth et al. Moist moss tundra on Kapp Linne, Svalbard is a net source of 1 CO2 and CH4 to the atmosphere.

Overall:

The ms addresses GHG measurements from a very remote part of the World and in the high arctic. As pointed out in the introduction few trustworthy measurements are available from the high Artic area as a whole and especially full year estimates and measurements are rare. The ms is written in an unpretentious way and aim to fill a piece of the GHG puzzle of the Arctic ecosystems, where also seasonal observations can be appreciated. As such measurements from the summer (late June – mid September) 2015 are reported on and also used for an interpretation of the annual CO2 /CH4 GHG budget of this tundra landscape.

So far so good. When I despite this feel that the authors still need to do some work with this ms it is because I find that especially the interpretation is take a bit further than the data can justify. Even if several assumptions are made (which is fine) it is not clear to the reader why some of the figures look like they do or why e.g. there are uncertainty of some definitions (e.g. growing/summer season) which I find confusing and suboptimal for the further interpretation of the data. I was first puzzled by the way data was presented in fig. 8 where supposing half hourly observations are compiled into a bi-weekly plot, leaving a peculiar impression of the dynamics of summer season in this ecosystem. It also leaves you with the impression that the peak of the summer is missing. However, it was not until I downloaded the data (which the general reader will not do) that I discovered that a whole month of data from later July to late August is completely missing in the flux observations. This should be made very clear in text as well as a general improvement of the description of the data coverage, which is not apparent from the figures and is critical for the further interpretation of the data. Based on this it is difficult for the reader to have confidence in the annual flux estimates (despite assumptions made) when also the peak of summer is missing in observations.

It would therefore be my recommendation that the revised manuscript relied more on the direct comparison of the measured data and the observed balance in GHG effect during the summer season, and made less attempts to extrapolate these into annual values, which it is difficult to have any confidence in. The observed data are rare and look fine (those which are there) and the authors could avoid a lot of the uncertainty by having a focus on that and a more direct comparison with other seasonal measurements from the Arctic.

[Figure]

Specific:

L22: the numbers seems to contradict the conclusion L535

L74: other studies could be mentioned e.g. Jammet 2017- Year-round CH4 and CO2 flux dynamics in two contrasting freshwater ecosystems of the subarctic, Biogeosciences, 14, 5189–5216, 2017 https://doi.org/10.5194/bg-14-5189-2017

L117: the hypothesis seems unargued and not very helpful, e.g. what is average CH4 flux ? please strengthen

L187: Some of these observations are quite far away, please reflect on the impact of that.

L291: I find the definitions of the seasons difficult to relate precisely to e.g. what is "*daily air temperature started to stay above zero more steadily*" or "*when most of the snow had disappeared*" – quantified how?

L330: a further sub-division of the vaguely defined season does not help me. Please consider a simplification.

Table 1: May all be significant but none of the three explain much of the variance – please explain

L367: No permafrost here???

Figure 8: as indicated earlier I find the time steps and general impression of the seasonal variability off and not a good representation of the actual measurements.

Figure 9: I have difficulties with this one as well both because of the negative GPP values – negative photosynthesis? Which appears because GPP is normalized for light response. I don't think the you can do that with two parameters which as dependent as temperature and light.

---

## Author Comment (AC1)

**RC2**: 'Comment on bg-2021-308', Anonymous Referee #2, 28 Jan 2022 reply

The study by Lindroth et al. characterises growing season net CO2 and CH4 fluxes in a tundra environment on Svalbard and attempts to extrapolate the growing season measurements to estimate annual fluxes. The manuscript certainly addresses an important research question related to possible climate change effects on greenhouse gas fluxes in the Arctic and provides important direct measurements of these fluxes. However, in my opinion, the methodology used in this study is not appropriate to answer these research questions. Extrapolating growing season fluxes to winter fluxes using a universal functional relationship with temperature risks causing major biases in annual fluxes. Ecosystem respiration function parameters can be expected to change between seasons and should not be assumed to be constant. Therefore, in my opinion, the main results (estimates of annual fluxes) of this study are not robust. Additionally, the use of a global warming potential to compare climate impacts of net CH4 and CO2 emissions is not justified for continuous greenhouse gas emissions in ecosystems. Other approaches should be considered in this case (see Neubauer, *Ecosystems*, 2021 or Neubauer & Megonigal, *Ecosystems*, 2015).

*Answer: We agree that the winter extrapolation is uncertain and that a focus on growing season is better. Thanks to this comment we discovered an error in the spread sheet calculation of growing season estimate and since we also adjusted the summer period a little, our numbers changed for both for summer and for growing season. We include a new Table 2 with the correct numbers (see below).*

*We also decided not to use the GWP concept and we only present the actual measured values.*

Table 2. Summary of seasonal C-fluxes from Kapp Linne.

| Period | Component | Value (gC m$^{-2}$) |
|---|---|---|
| Growing season | Reco | 110.2 |
| | GPP | -112.7 |
| | NEE | -2.5 |
| Summer | Reco | 94.1 |
| | GPP | -105.9 |
| | NEE | -11.8 |

---

## Author Comment (AC2)

**RC1**: 'Comment on bg-2021-308', Thomas Friborg, 27 Jan 2022 reply

Lindroth et al. Moist moss tundra on Kapp Linne, Svalbard is a net source of 1 CO2 and CH4 to the atmosphere.

Overall:

The ms addresses GHG measurements from a very remote part of the World and in the high arctic. As pointed out in the introduction few trustworthy measurements are available from the high Artic area as a whole and especially full year estimates and measurements are rare. The ms is written in an unpretentious way and aim to fill a piece of the GHG puzzle of the Arctic ecosystems, where also seasonal observations can be appreciated. As such measurements from the summer (late June – mid September) 2015 are reported on and also used for an interpretation of the annual CO2 /CH4 GHG budget of this tundra landscape.

So far so good. When I despite this feel that the authors still need to do some work with this ms it is because I find that especially the interpretation is take a bit further than the data can justify. Even if several assumptions are made (which is fine) it is not clear to the reader why some of the figures look like they do or why e.g. there are uncertainty of some definitions (e.g. growing/summer season) which I find confusing and suboptimal for the further interpretation of the data. I was first puzzled by the way data was presented in fig. 8 where supposing half hourly observations are compiled into a bi-weekly plot, leaving a peculiar impression of the dynamics of summer season in this ecosystem. It also leaves you with the impression that the peak of the summer is missing. However, it was not until I downloaded the data (which the general reader will not do) that I discovered that a whole month of data from later July to late August is completely missing in the flux observations. This should be made very clear in text as well as a general improvement of the description of the data coverage, which is not apparent from the figures and is critical for the further interpretation of the data. Based on this it is difficult for the reader to have confidence in the annual flux estimates (despite assumptions made) when also the peak of summer is missing in observations.

*Answer:*

*1. The reason why we chose to have two different definitions of growing season is because there is no standardized definition of growing season. We mention this in L 503. This is illustrated by e.g Oechel et al. (2014) which used five different definitions of growing season. But we also agree that we could be more explicit in our formulations and we suggest the following changes: We retain only one definition of growing season, the one based on permanence of snow cover and we are more explicit in the definition of 'summer' period which now does not include the first 8 days in June where there was a snow cover. Thus, summer is here defined as the period from 9 June to 31 August. Accordingly, we also changed Figure 2 (see below).*

[Figure]

2. *Concerning flux observations, we agree that we should have given more details about data coverage. However, the gapfilling of the missing data is made with a well-established and thoroughly tested tool (Wutzler et al. 2018) which is commonly used by the flux community and which we have confidence in. We added the following ". The total data coverage during this period was 47% with a longer break in the measurements between 28 July and 29 August."*

3. *Regarding comment about 'missing peak of summer season'. We agree that Fig. 8 does not give a good description of the dynamics of the system and we therefore suggest to remove Fig. 8 and instead show a figure of the diurnal course of NEE during each of the months June-September (new Fig. 8, see below). We also suggest to add a Fig. 9 showing the cumulative NEE during the growing season. It is obvious from these new figures that we did not 'miss' the peak of the summer season but that the high respiration and the relatively low GPP resulted in a very flat behavior of the season course (previous Fig. 8) when averaging was made over several 24 h cycles.*

[Figure]

*Figure 8. The mean monthly diurnal course of NEE during the period of eddy covariance measurements 25 June to 17 September. The error bars (only every 2nd is shown) are the 95% confidence interval.*

[Figure]

*Figure 9. The cumulated half-hourly NEE during growing season.*

It would therefore be my recommendation that the revised manuscript relied more on the direct comparison of the measured data and the observed balance in GHG effect during the summer season, and made less attempts to extrapolate these into annual values, which it is difficult to have any confidence in. The observed data are rare and look fine (those which are there) and the authors could avoid a lot of the uncertainty by having a focus on that and a more direct comparison with other seasonal measurements from the Arctic.

*Answer: We agree that the winter extrapolation is uncertain and that a focus on growing season is better. Thanks to this comment we discovered an error in the spread sheet calculation of growing season estimate and since we also adjusted the summer period a little, our numbers changed for both for summer and for growing season. We include a new Table 2 with the correct numbers (see below).*

*Table 2. Summary of seasonal C-fluxes from Kapp Linne.*

| Period | Component | Value (gC m$^{-2}$) |
|--------|-----------|---------------------|
| Growing season | Reco | 110.2 |
| | GPP | -112.7 |
| | NEE | -2.5 |
| Summer | Reco | 94.1 |
| | GPP | -105.9 |
| | NEE | -11.8 |

Specific:

L22: the numbers seems to contradict the conclusion L535

*Answer: We have skipped the calculation of GWP values based on recommendation from another reviewer.*

L74: other studies could be mentioned e.g. Jammet 2017- Year-round CH4 and CO2 flux dynamics in two contrasting freshwater ecosystems of the subarctic, Biogeosciences, 14, 5189–5216, 2017 https://doi.org/10.5194/bg-14-5189-2017

*Answer: Thanks, we have added a reference but choose Jammet et al. (2015) instead.*

*Jammet, M., Crill, P., Dengel, S. and Friborg, T.: Large methane emissions from a subarctic lake during spring thaw: Mechanisms and landscape significance. J. Geophys. Res.-Biogeo., 120, 2289-2305, http://doi.org/10.1002/2015JG003137, 2015.*

L117: the hypothesis seems unargued and not very helpful, e.g. what is average CH4 flux ? please strengthen

*Answer: We changed the text to 'We hypothesise that this moist tundra ecosystem is a net carbon sink during the growing season and that the summer emissions of methane will be at levels comparable with other methane emitting high Arctic ecosystems.'*

L187: Some of these observations are quite far away, please reflect on the impact of that.

*Answer: We will add the following text to that section: Using data from the more distant locations, Svalbard airport and Adventdalen, introduces some additional uncertainty. Concerning global radiation data we could compare in situ measured half-hourly radiation with the corresponding data from Adventdalen for a shorter period and it showed general good agreement although with relatively large scatter (y = 0.84x + 15.9; $r^2$=0.57; n=580). According to Dobler et al. (2020) the amount of precipitation in the area where Kapp Linne and Svalbard airport are located don't show any significant differences on an annual basis and Vickers et al. (2020) analysed timing of snow cover in Svalbard and they show that the mean (2000-2019) first snow-free day is very similar in areas where Kapp Linne and Svalbard airport are located. Thus, we are confident that using data from these relatively remote locations does not introduce serious bias in our analyses.*

L291: I find the definitions of the seasons difficult to relate precisely to e.g. what is "*daily air temperature started to stay above zero more steadily*" or "*when most of the snow had disappeared*" – quantified how?

*Answer: We have removed the temperature dependent definition as described above and we try to be more precise in definition of the growing season as described above.*

L330: a further sub-division of the vaguely defined season does not help me. Please consider a simplification.

*Answer: Good point, we have changed the text to specific dates.*

Table 1: May all be significant but none of the three explain much of the variance – please explain

*Answer: We don't see any other reason than that there is a large scatter in the data as shown in Fig. 4 and that there are several variables that affect the $CO_2$ fluxes.*

L367: No permafrost here???

*Answer: This whole paragraph (L362-L369) is removed since we focus on growing season.*

Figure 8: as indicated earlier I find the time steps and general impression of the seasonal variability off and not a good representation of the actual measurements.

*Answer: See above, Fig. 8 replaced by new plus a new Fig. 9.*

Figure 9: I have difficulties with this one as well both because of the negative GPP values – negative photosynthesis? Which appears because GPP is normalized for light response. I don't think the you can do that with two parameters which as dependent as temperature and light.

*Answer: It is a common procedure to estimate a 'light response curve' from measurements of the net exchange without considering effects of temperature. We believe that such curves truly show the overall light response and that the temperature response is shown by the scatter around the light response curve. In this paper we have used the sign convention that a positive flux is upwards from the ecosystem into the atmosphere and vice versa for a negative flux. Thus, negative GPP means uptake. But as always when working with EC measurements there is a scatter in the data because of the nature of turbulent exchange (see e.g. Fig. 7). Thus, getting both positive and negative values in a situation like this is not unusual. It is the overall response that is important and not the individual points. If we are convinced that the light response curve reflect the impact of light on the processes, then the method that we have used should be in order.*

---

## Author Response (AR1)

Comments to the Editor:

We have included all changes as mentioned in our reply to reviewer 1 and 2 and we don't repeat this here.

**Comments to the author:**

This is an intriguing dataset of CO2 and CH4 fluxes in a very remote part of the world. Both referees are supportive of publishing the work, but both have concerns. Please address those concerns as you have suggested in a revised version of the paper.

I share the concerns about extrapolating to the full year from a few months in summer - please remove that content and focus on the time period of measurements.

**Answer: Done. Changes have been done in many places, please see the annotated version.**

I agree that the NEE partitioning methods are well accepted. (Citing Wutzler et al (2018) is not enough because the REddyProc package includes both the nighttime (Reichstein et al. 2005) and daytime (Lasslop et al. 2010) partitioning methods. Presumably you are using the daytime method, be clear about this.). However, you are missing data from most of the month of August. Your new Fig 8 which shows the monthly diel patterns highlights that the missing month is actually during the transition from peak growing season to full respiration in September. Please add something to the August panel that clearly indicates on the figure that the pattern results from all/mostly gap-filled data, and some text that highlights the limitations.

Answer: Thanks for pointing this out. We are using the daytime method and we have added a reference to Lasslop et al in the text L225-226. A comment about the missing data is also made in the panel of Fig. 8 and we also made a remark in the text L415-420 following Fig. 8.

I am not concerned with presenting GPP as a negative number (Fig 7) as long as you are clear about the sign convention. I am however confused about the focus on temperature sensitivity of GPP in Figure 9. The application of the Lasslop method a priori assumes, and thus provides, an exponential response of ecosystem respiration to temperature. The GPP term is modeled strictly as a response to radiation (or also humidity as shown by Lasslop). Noise around the GPP light response fit line is caused by many factors, perhaps a temperature effect on GPP as you suggest, but also by changes in stationarity of turbulence, the low u\* underestimation of fluxes, time-varying footprint changes, etc.

I do not agree that the residual of the GPP-light relationship can be interpreted as the temperature sensitivity of GPP. Your new Figure 8 highlights the very strong seasonal change in GPP during this short growing season. Plant photosynthesis is responding to seasonal change in temperature in a dramatic way in your dataset. I suggest that you evaluate the GPP-light response in successive 2-weekly periods. I suspect you will find a change in the light-saturated value of GPP during those different periods (analogous to seasonal change in leaf-level photosynthetic capacity). For example, Figure 5 in Bowling et al. (2018) illustrates these patterns during the spring transition from winter dormancy to active photosynthesis in a forest. That change in my opinion is a more appropriate way to examine how plant photosynthesis is changing as temperature changes across the season. I don't insist that you make these changes, but please consider them and modify the paper accordingly if you find this useful.

Answer: Referring to our mail conversation we have added some text about the analyses of which variables (Ta, time of season, vpd) that best explain the the variance of the normalized GPP (L 459-463). We did not include the reference below since we didi not use this method.

Bowling, D. R., Logan, B. A., Hufkens, K., Aubrecht, D. M., Richardson, A. D., Burns, S. P., Anderegg, W. R. L., Blanken, P. D., and Eiriksson, D. P.: Limitations to winter and spring photosynthesis of a Rocky Mountain subalpine forest, Agricultural and Forest Meteorology, 252, 241–255, https://doi.org/10.1016/j.agrformet.2018.01.025, 2018.

---

## Author Response (AR2)

Dear Experienced Colleagues,
As indicated in my previous comments I can appreciate the observations from a remote location, and would gladly see you publish the measurements that you actually have. What I do not like so much is that you make a few measurements look like they covered a full month or summer or year - we're no longer in year 2000 where you can measure a few days and just guess the rest. We're in the era of monitoring programs and standardized measurement, where quality of the measurements is essential. I recommend that you cut out any extrapolations where you have less than 50% of measured data and present the rest as observations from an extreme climatic location. Then the modelers can do the modelling. For me this is a weird mix that claims to be observations.

*We appreciate that the reviewer appreciate our observations from a very remote place on Earth and that he is glad (partly) to see our results published. However, he does not like the gap filling, particularly the long period in August with missing observations. But calling gap filling for 'guesses' is purely nonsense and either the reviewer does not know how the Wutzler et al gap filling works or he just dislike gap filling in general. But maybe it is more related to his comment about us being in "the era of monitoring programs and standardized measurements, where quality of the measurements is essential". For sure quality of measurements are essential but this does not only hold for scientists who are lucky to work in such circumstances, quality is always essential. And we must stress that we have never claimed that the "weird mix" are observations only. But we have tried to make this issue even more visible for instance see L 157-158 and L 222-225 in the revised version.*

L: 17: I still don't think it can be justified to estimate summer fluxes or August fluxes based on two day only. Leave August out, and account for June and July, otherwise this adds more to confusion than conclusion.

*We don't agree. We assume that a serious reader not only read the abstract but also look in the main text and there it is discussed the issue of gapfilling. Besides the clarifications mentioned above we have also added a paragraph L 408-412 with reference to the supplement where we show a comparison between gap filled and modelled diurnal courses. We believe that most readers will appreciate our attempt to estimate the season fluxes and not only show bits and pieces with maximum data coverage.*

Figure 4 what is on the x axis?

*It says in the x-axis legend 'Sampling occasion' and we don't think it needs further explanation.*

Figure 6. Measured ecosystem respiration (Reco; green dots)- in ½ hourly means?

*We added the word 'using chambers' in the legend to make sure that the reader understand from where these measurements come.*

Figure 8: is the diurnal cycle in august based on the two last days of the month only? And June on the last 6 days only ? you should then say "Notice that the main part of June and August was gap filled" What about sep? do you have more than 50% data coverage here? Otherwise include that in the sentence as well.

*It is clear in the figure legend that the diurnal courses are based on EC measurements during the period 25 June to 17 September and not the full growing season. We already added a notice about the long gap filled period in August.*

L: 410 measurement problems.

*Don't understand what the reviewer mean with this comment?*

Figure 9: to me that looks like the previous ms version fig.8 , which I didn't like, and still don't, because there are no measurements from August and it still looks weird to me. Maybe worse now because it looks like numbers were measured rather than guessed

*Maybe the reviewer don't like this figure but we don't get any explanation why it is 'weired'. What is 'weired' with this curve? And again, a reader who read the main text will for sure understand that this curve is based on both measured, gap filled and modelled (at end and beginning of season – clearly stated in main text) data. How can it 'look like' the numbers are measured instead of guessed?*